# First Evaluation of a New Dynamic Scoring System Intended to Support Prescription of Adjuvant CytoSorb Hemoadsorption Therapy in Patients with Septic Shock

**DOI:** 10.3390/jcm10132939

**Published:** 2021-06-30

**Authors:** Klaus Kogelmann, Tobias Hübner, Franz Schwameis, Matthias Drüner, Morten Scheller, Dominik Jarczak

**Affiliations:** 1Department of Anesthesiology and Intensive Care Medicine, Klinikum Emden, Bolardusstr. 20, 26721 Emden, Germany; m.druener@klinikum-emden.de (M.D.); m.scheller@klinikum-emden.de (M.S.); 2Department of Anesthesiology and Intensive Care, Kantonsspital Münsterlingen, Spitalcampus 1, 8596 Münsterlingen, Switzerland; tobias.huebner@stgag.ch; 3Department of Anesthesiology and Intensive Care, Landesklinikum Baden-Mödling, Sr. M. Restituta-Gasse 12, 2340 Mödling, Austria; office@schwameis.com; 4Department of Intensive Care, Universitätsklinikum Hamburg Eppendorf, Martinistr. 52, 20251 Hamburg, Germany; d.jarczak@uke.de

**Keywords:** inflammation, septic shock, CytoSorb, hemoadsorption, scoring

## Abstract

Introduction: Despite advances in critical care medicine, adjunctive approaches in sepsis therapy have failed to prove their efficacy. Notwithstanding promising results using hemoadsorption (CytoSorb), questions remain concerning timing and dosing. We created a dynamic scoring system (DSS) to assess patients with early septic shock and performed a first evaluation of the system in this patient population. Methods: Data from 502 patients with septic shock according to Sepsis-3 criteria were retrospectively analyzed. Score parameters were documented at the time of diagnosis (T_0_) and 6 h later (T_6_) to calculate a dynamic score. Survival on day 7 and 56 as well as ICU and hospital mortality were analyzed in regard to the score as well as the delay of hemoadsorption therapy. Results: Of the 502 patients analyzed, 198 received adjunctive CytoSorb treatment and 304 received standard therapy. Septic shock was typically represented by 5 points, while >6 points indicated a situation refractory to standard therapy with the worst outcome in patients shown by >8 points. The differences in mortality between the score groups (<6, 6–8, >8 points) were significant. Analysis further showed a significant 56-day, ICU and hospital survival advantage in CytoSorb patients when therapy was started early. Conclusion: We created a scoring system allowing for the assessment of the clinical development of patients in the early phase of septic shock. Applying this approach, we were able to detect populations with a distinct mortality pattern. The data also showed that an early start of CytoSorb therapy was associated with significantly improved survival. As a next step, this easy-to-apply scoring system would require validation in a prospective manner to learn whether patients to be treated with hemoadsorption therapy in the course of septic shock could thereby be identified.

## 1. Background

Sepsis represents a major challenge for medicine and a significant public health concern [1]. Despite all medical advances in recent years, it continues to be a substantial problem, as to date therapeutic approaches have failed to prove efficacy [2]. Sepsis has major importance from a medical and from an economical viewpoint. Approaches that could contribute to its successful treatment need to be further explored [3].

In recent years, hemoadsorption (CytoSorb) has been used more and more frequently to treat septic shock, especially in refractory conditions, in which standard therapy did not seem to be sufficient enough [4,5]. The exact state of ‘refractory shock’, however, is not well defined [6]. As there are several lines of evidence showing that an early start of hemoadsorption therapy might be beneficial [4,5,7,8], start of such an approach should not be delayed too long to preserve chances of success. The primary clinical effect of CytoSorb therapy is reported to be a stabilization in hemodynamics accompanied by the opportunity to decrease catecholamine dosages [4,5,6,7]. This has been shown to go along with a concomitant restoration in metabolic parameters [9,10]. It therefore appears reasonable to also use hemodynamic (and metabolic) parameters in order to define better criteria for therapy initiation but also to shed more light on the definition of ‘refractory shock’. For this reason, a dynamic scoring system (DSS) was created based on established, clinically well-available and hemodynamics-associated parameters such as lactate, volume and catecholamine therapy and their changes within the first 6 h. This system should allow for the assessment of the early phase of septic shock to better define refractory states and finally—to be completed in upcoming, prospective analysis still to be performed—help to identify patients with refractory septic shock, who might benefit most from adjuvant therapy with CytoSorb hemoadsorption, early. As a first step, the dynamic system was evaluated via retrospective data analysis of 502 patients with septic shock, 198 of which had been treated with adjunctive hemoadsorption therapy. Additionally, the impact of therapy delay in regard to the initiation of hemoadsorption was evaluated.

## 2. Material and Methods

### 2.1. Ethics Approval, Legal Considerations

This study was approved by the ethics committee of the General Medical Council of Lower Saxony (reference number Bo/29/2019). The study was carried out according to the Declaration of Helsinki and in accordance with the Good Clinical Practice Protocol (GCP) (2001/20/EEC, CPMP/ICH/135/95), the established standard operating procedures and the local laws and regulations applicable to each country. The study was registered at ClinicalTrials.gov, 6 June 2019 (NCT03977688).

### 2.2. Study Design

This study was a retrospective data analysis in 502 critically ill adult patients. Included were data from 4 interdisciplinary intensive care units (ICU) with comparable procedures (Emden/Germany, Münsterlingen/Switzerland, UKE Hamburg/Germany, Baden-Moedling/Austria). Inclusion criteria comprised coded diagnosis of septic shock (Sepsis-3 criteria) [1]. Septic shock is defined according to the SCCM/ESICM Sepsis-3 definition [1], i.e., vasopressor requirement to maintain a mean arterial pressure of 65 mmHg and serum lactate level >2 mmol/L in the absence of hypovolemia. We excluded patients where data records were unavailable for analysis, patients not treated in the ICU and patients where norepinephrine (NE) requirement or lactate were not documented.

### 2.3. Objectives

Survival on day 56 in regard to DSS score was defined as the primary objective. Secondary objectives included survival on day 7, ICU and hospital mortality, as well as the timing of hemoadsorption therapy, catecholamine demand, lactate, inflammatory parameters (PCT, CRP), creatinine, duration of organ support (ventilation, renal replacement therapy (RRT), CytoSorb therapy) and length of stay in ICU and hospital.

### 2.4. Assessed Parameters

The following parameters were assessed: medical history, patient characteristics, disease severity scores (Acute Physiology and Chronic Health Evaluation II—APACHE II, Simplified Acute Physiology Score 2—SAPS 2), hemodynamics (catecholamine demand, heart rate, blood pressure), laboratory parameters (lactate clearance, inflammatory parameters, creatinine), initial volume requirement to achieve intravascular normovolemia, use of either hydrocortisone or a second catecholamine (or both), CytoSorb-therapy specific data (therapy delay after diagnosis of septic shock), duration of organ support (duration of mechanical ventilation, renal replacement therapy and CytoSorb therapy), outcome data (day 7 and day 56 survival, ICU and hospital stay and survival) as well as safety relevant issues (adverse events).

### 2.5. Data Collection

Data were stored in the hospital information system and could only be accessed by the investigator in charge. Participating investigators provided their data pseudonymized in a tabular format. Centralized data processing was performed at the Department of Anesthesiology and Intensive Care at Emden Hospital.

### 2.6. Procedure

Collected data were entered into a data matrix, presenting the created dynamic scoring system (Figure 1). Each parameter at the time of septic shock diagnosis (T_0h_) and 6 h (T_6h_) later was documented to analyze the initial status as well as the dynamic process in early septic shock. We decided to use the interval of the first 6 h for our dynamic scoring, as this initial course, which is also targeted by the Sepsis Bundles [11], might play an even more important role than changes over several days. The threshold values are based on Sepsis-3 criteria of septic shock [1] (lactate level >2 mmol/L), the administration of norepinephrine according to the Sequential Organ Failure Assessment (SOFA) Score [12] (norepinephrine > 0.1 μg/kg/min) and volume requirement according to the Surviving Sepsis Guidelines [11] (bolus 30 mL/kg body weight).

Each pathological value at time of diagnosis (T_oh_) was rated with 2 points. If <2 volume boluses of 30 mL/kg were necessary, this was rated with 1 point, whereas ≥2 volume boluses of 30 mL/kg within 6 h were then scored with 2 points. Further changes after 6 h (T_6h_) were rated as follows: decreasing values or no change during the observation period received no points, increasing values received 1 point and increasing values >50% received 2 points. (Figure 1). Using these data, we finally defined 3 different groups of patients (<6 points, 6–8 points, >8 points). Septic shock was represented by a score of 5 points (lactate ≥ 2 mmol/L, NE ≥ 0.1 µg/kg/min needed to maintain a mean arterial pressure (MAP) ≥65 mmHg and initial volume bonus of 30 mL/kg applied), while 6 or more points indicated a situation refractory to standard therapy with an even worse clinical course given in patients with >8 points.

### 2.7. Statistics

All primary and secondary variables were first examined using an exploratory data analysis method and recorded descriptively. Data are reported as mean ± standard deviation, frequency and percentage, or median as required. A normal distribution was tested using the Shapiro–Wilk test. Differences in the primary endpoint between study groups were analyzed using the Chi-square (χ^2^) test, using a Bonferroni correction for multiple comparisons (α/number of groups compared = adjusted critical value). To compare the survival function between cohorts, non-parametric Kaplan–Meyer survival analysis was performed on day 7 and 56, estimating and plotting the survival probability as a function of time. The different survival probabilities of groups were then compared to each other regarding statistical significance using a Log-rank test to compare groups. Secondary endpoints were tested with an independent sample *t*-test, or χ^2^ test, with Bonferroni correction, as required. For non-normal distribution results, nonparametric tests were performed. Data were analyzed with SPSS 20.0, a value of *p* < 0.05 was defined as the α (alpha) critical value (statistically significant).

## 3. Results

In the study, 502 patients were included, 61.8% of whom were male. A total of 198 patients were treated with CytoSorb (39.4%) and 304 received therapy without CytoSorb (60.6%). Diagnoses in the study population included pneumonia (*n* = 219, 43.6%), abdominal sepsis (*n* = 186, 37.1%), uro sepsis (*n* = 38, 7.6%) and miscellaneous (*n* = 59, 11.7%). The baseline characteristics are depicted in Table 1. With regard to gender, the APACHE 2 Score and ICU and hospital days, there were no significant differences between the groups. Significant differences were found in age, SAPS 2, ventilation days, ICU and hospital mortality, all of which scored items and points in the dynamic scoring system (Table 1). Analysis of the inflammatory parameters did not show any correlation between the score groups, apart from patients with a DSS < 6 points had the lowest PCT levels.

In the overall patient population, the primary endpoint analysis showed that higher DSS scores were associated with an increase in day 56 mortality (<6 vs. >8; *p* = 0.004) (Figure 2), when both ICU and hospital mortality are considered. Kaplan–Meier curves showing the effect at day 56 as well as day 7, are provided below (Figure 2 and Figure 3).

A total of 198 patients were treated with CRRT and CytoSorb, 61.1% of whom were male. The diagnoses in this study population included pneumonia (*n* = 76, 38.3%), abdominal sepsis (*n* = 85, 42.9%), urosepsis (*n* = 13, 6.5%) and miscellaneous (*n* = 24, 12.1%). The baseline characteristics are depicted in Table 2. With regard to gender, the APACHE 2 Score and ICU and hospital days as well as ICU-mortality, there were no differences between the groups. Significant differences were found in age, SAPS 2, ventilation days, hospital mortality and time delay until start of therapy, all of which scored items without T_0_ lactate and points in the dynamic scoring system (Table 2).

In this cohort of patients, who received CytoSorb treatment, those with a DSS < 6 points counterintuitively showed an increased ICU and hospital mortality when compared with clusters with 6–8 points. However, a comparison of the clusters with 6–8 points and >8 points again confirmed the trend of higher DSS scorings being linked to an increased mortality. Taking a closer look at the first 7 days, however, patients with a score <6 showed, as originally expected, a lower mortality compared to the other groups (23.5 vs. 24.4 vs. 38.7%) (Table 3). Therefore, apart from the above-mentioned exceptions, differences in mortality showed a trend with increasing score groups in the first 7 days, after 56 days and also with regard to ICU and hospital mortality (Table 2, Figure 4 and Figure 5).

To also investigate the association between mortality and the time until the start of hemoadsorption therapy in patients receiving CytoSorb, the delay from onset of septic shock (T_0_) to the initiation of hemoadsorption therapy was clustered into different time intervals (≤12 h, >12–24 h, >24 h). The time until the start of therapy correlated positively with ICU and hospital mortality (Table 4). After 7 days, however, this was not significant (29.1 vs. 31.8 vs. 34%), (Figure 6). After 56 days, mortality in the group with a 12-h delay was lower than in the group with >24-hour delay (Figure 7).

Mortality was lowest in the patients where therapy was started early, despite the patients having higher lactate levels and norepinephrine needs and a higher need for a second catecholamine and/or hydrocortisone. This was significant for ICU mortality at ≤12 vs. >24 h therapy delay (50 vs. 73.5%, *p* = 0.006), for hospital mortality when comparing ≤12 vs. >24 h therapy delay (55 vs. 79.2%, *p* = 0.004) and for 12–24 h vs. >24 h therapy delay (61.5 vs. 79.2%, *p* = 0.038) (Table 3).

The mean therapy delay for the defined time intervals was significantly different (6.8, 20.4 and 54.6 h, respectively; *p* < 0.001). The patients in whom therapy was started earliest, had the highest DSS score (8.4 vs. 7.6 vs. 7.0 DSS points), while patients with a low DSS had a longer delay until the start of CytoSorb (Table 2 and Table 4). For better comparability of patients according to their severity of illness, we analyzed the subgroup of septic patients who did not receive CytoSorb therapy but who had ARF with a need for CRRT (n = 69), especially as all the CytoSorb-treated patients had also received concomitant CRRT due to ARF. The baseline characteristics are depicted in Table 5. With regard to gender, APACHE 2 Score, ventilation and ICU and hospital days, there were no differences between these patients in comparison with the CytoSorb patients group. ICU and hospital mortality in the CytoSorb-treated patients were lower, but not significant (Table 5).

In the CytoSorb group, a second catecholamine (56 vs. 24.6%) as well as hydrocortisone (69.1 vs. 53.6%) were used more frequently. The same held true for norepinephrine demand, which was also significantly increased at T_0_ and T_6_ in this group and, therefore, suggests that patients in the CytoSorb group had an increased disease severity. If therapy delay is combined with the disease severity (determined by the DSS score), the mortality of CytoSorb patients is lower in the group with 6–8 points and a therapy delay ≤12 h, becoming significant in the group with >8 points (Table 6) for ICU mortality, after a Bonferroni adjustment.

A multivariate logistic regression model was fit to investigate an association between day 56 survival and selected predictor variables (Table 7). The results showed that the use of the CytoSorb device reduced the odds of mortality at day 56 by 44.8%. With regard to the DSS Score, for each one unit increase in score, the odds of mortality at day 56 increased by 23.7% (*p* <0.001). Similarly, for each additional hour in CytoSorb therapy delay, the odds of mortality at day 56 increased by 1.5% (*p* = 0.034); the associated use of renal replacement therapy (RRT) increased the odds of mortality at day 56 by 75.9%; and lastly, for each one-year increase in patient age, the odds of mortality at day 56 increased by 3.7% (*p* <0.001).

## 4. Discussion

In this retrospective, non-interventional, two-arm, multicentric data analysis, a newly created dynamic assessment system based on the clustering of established, clinically well-available and predominantly hemodynamics-associated parameters was used and evaluated in regard to its association with mortality. In the overall patient population (Table 1, Figure 2 and Figure 3), a higher DSS score was associated with increased mortality at day 56, supporting the validity of the established procedure and analysis of data, but also the impact and predictivity of the assessed variables and their dynamic change on the outcome. In regard to the impact of the timing of hemoadsorption therapy, earlier initiation was shown to be associated with a better outcome.

With one exception, the correlation between higher scores and mortality was the same in the CytoSorb-treated patient population (Table 2, Figure 4 and Figure 5), as CytoSorb treated patients with a DSS <6 failed to show this correlation. Similar findings were observed by Ferreira and colleagues in an analysis of the outcomes related to changes in the SOFA score [13]. The authors found an increased mortality in a group with a decrease in SOFA score points and presumed decreasing mortality, which was ultimately explained by the small number of patients (7.5% of the total group). This was also exactly the case in our set of patients (*n* = 17, 8.5% of the total CytoSorb group). Moreover, taking into consideration the fact that timing seems to be a very important aspect of the therapy with impact on outcome, the delay time until the start of hemoadsorption therapy was longer in this small subgroup (52.6 h) compared to all other groups. Both can help to explain the higher mortality rate in these patients. In contrast to established scoring methods such as APACHE 2, SAPS 2 and SOFA, the score shown here does not describe a course over 24 h but can rather be used at the bedside within a very short time and, thus, describes early septic shock [13,14,15,16]. We performed survival analyses for the first 7 days and at day 56. Focusing on only the first 7 days in some analysis can be explained by the fact that the start of CytoSorb therapy is followed by an average of three days of therapy [17]; therefore, the maximum effect of this therapy is most likely to occur after day 3 and is potentially most pronounced in the first few days and not necessarily in the later clinical course [4,5,7,9,10]. However, there are also reports of differences in the outcome only observed at a later stage of the clinical course [18].

Several recent articles on the use of CytoSorb therapy have highlighted the potential benefits of an early start of hemoadsorption treatment [4,5,7], which led us to analyze the correlation between the time delay until therapy initiation and mortality between the groups (Table 4, Figure 6 and Figure 7). The mean therapy delay for the time intervals was significantly different between the DSS groups (6.8, 20.4 and 54.6 h, respectively; *p* < 0.001). A delayed initiation of hemoadsorption therapy was strongly associated with higher day-56, ICU and hospital mortality (Table 4), even though the patients in which CytoSorb therapy was started earlier, had higher norepinephrine requirements, an additional need for a second catecholamine as well as hydrocortisone and a higher DSS score (Table 2). Therefore, despite these patients being sicker than those treated later, they had a significantly better outcome, which further supports the assumption that timing as well as actual patient status are important criteria to consider in regard to the initiation of hemoadsorption therapy.

Acute renal failure in sepsis is common [19,20] and associated with high mortality [21,22]. ARF in sepsis has a high mortality rate of more than 70% [23,24]. A SepNet prevalence study from Germany showed a significantly higher mortality rate in septic patients with ARF compared to those without ARF (67.3 vs. 42.8%) and concluded that ARF represents an independent risk factor for poor outcome in septic shock [25]. To shine a light on the role of acute renal failure and due to the fact that the CytoSorb-treated patients all received CRRT, we analyzed the subgroup of ECSISS patients, who did not receive CytoSorb therapy but who had also ARF with the need for CRRT (*n* = 69). Brouwer et al. showed that CytoSorb therapy was associated with a decreased observed versus expected 28-day all-cause mortality and may be associated with a decreased all-cause mortality at 28 days compared to CRRT alone [26]. Recent results from Rugg et al. even showed an observed significant mortality difference between septic shock patients treated only with CRRT versus those treated with CRRT and adjunctive CytoSorb therapy [18]. Overall, our data suggest that the need for RRT is strongly associated with mortality (Table 5 and Table 6). If the therapy delay for CytoSorb is analyzed together with the disease severity (determined by the DSS score) in regard to mortality and then compared to non-CytoSorb treated CRRT patients of the same DSS group, the mortality of CytoSorb patients with a therapy delay ≤12 h is lower in the DSS group of 6–8 points, becoming significant for ICU mortality in the group with >8 points (Table 6), a finding that further supports the recent results on the potential outcome benefits of CytoSorb therapy in patients with septic shock requiring CRRT. Finally, further analysis was performed to investigate the association of certain variables such as DSS score and therapy delay with mortality at day 56 via a multivariate logistic regression model (Table 7). The results support our initial findings and could even show some principle impacts of CytoSorb therapy on mortality, which, however, was not significant and also not the primary target of this investigation.

In our work, the severity of the disease in early septic shock is represented by the DSS score. Thus, more catecholamine requirements, the use of hydrocortisone and a second catecholamine as well as volume application and lactate plasma concentrations might represent disease severity better than 24-h scores such as APACHE 2, SOFA or SAPS 2 systemically can [13,14,15,16].

### Limitations

This study was performed as a retrospective data collection and is not a prospective randomized, controlled trial. The evaluation may include patients in whom an escalation of therapy has been avoided. This is less likely in the CytoSorb group, since an additional procedure was already in place. There were a number of patients with a ‘do not resuscitate’ (DNR) order in the data sets, which were consequently not included. An influence of country- or clinic-specific therapy regimes cannot be ruled out. By weighting the therapy measures (application of volume, catecholamines, hydrocortisone; Figure 1) differently in the DSS matrix, we have tried to compensate for this. Last but not least, the created scoring system has yet to be validated in a prospective approach in regard to how it could help to identify patients that are likely to benefit from CytoSorb therapy in septic shock.

## 5. Conclusions

This newly created dynamic score allows for the assessment of hemodynamic development in the early phase of septic shock, thereby detecting the refractory status of septic patients and, finally, differentiating them into subgroups with different mortalities. This was given for all the patients examined and also for those who had received adjunctive CytoSorb therapy. Additionally, we could again show that the earlier CytoSorb therapy was started, the better the outcome was in terms of mortality. This easy-to-apply scoring system, which requiring only classical clinical hemodynamic information and one further laboratory value (lactate clearance), might present an option to better detect patients benefitting from the initiation of hemoadsorption therapy in septic shock, but prospective validation in this regard is required first.

## Figures and Tables

**Figure 1 jcm-10-02939-f001:**
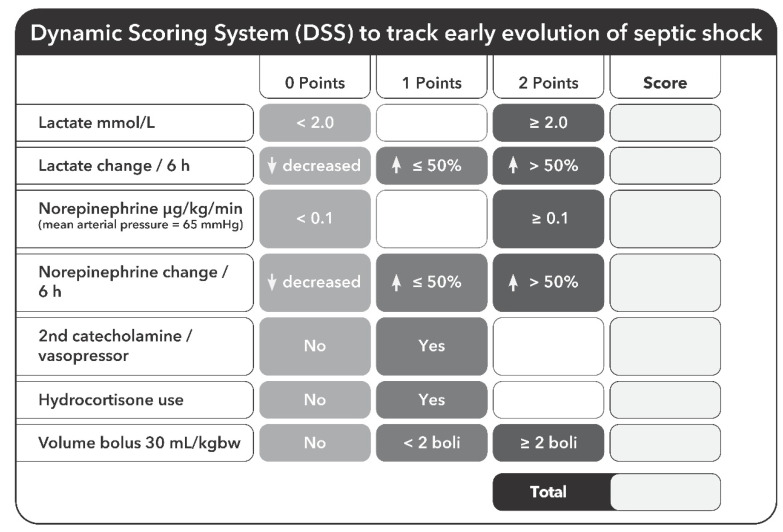
Dynamic Scoring System. Each parameter at the time of septic shock diagnosis (T_0h_) and 6 h (T_6h_) later is documented to analyze the dynamic process in early septic shock. Each pathological value is rated with 2 points. Decreasing values or no change during the 6 h observation period receives no points, increasing values receive 1 point, increasing values >50% of the initial value receive 2 points. If <2 volume boluses of 30 mL/kg are necessary, this is rated with 1 point, when ≥2 volume boluses of 30 mL/kg are necessary within 6 h, this is scored with 2 points.

**Figure 2 jcm-10-02939-f002:**
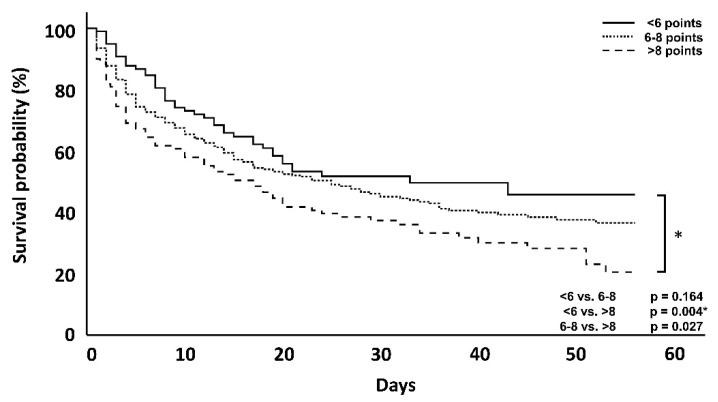
Kaplan–Meyer survival analysis at day 56 of the entire patient cohort. The *p*-values are via log-rank tests. * = Statistically significant using Bonferroni-adjusted alpha critical value = 0.017 for groups <6 vs. >8, but no other comparisons.

**Figure 3 jcm-10-02939-f003:**
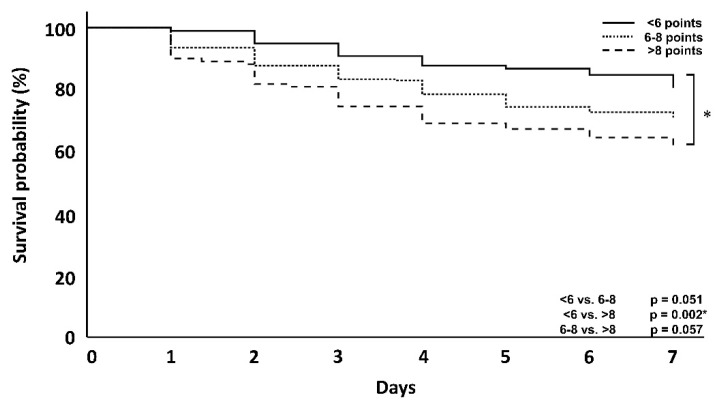
Kaplan–Meyer survival analysis at day 7 of the entire patient cohort. The *p* = value is via log-rank test. * = Statistically significant using Bonferroni-adjusted alpha critical value = 0.017 for groups <6 vs. >8, but no other comparisons.

**Figure 4 jcm-10-02939-f004:**
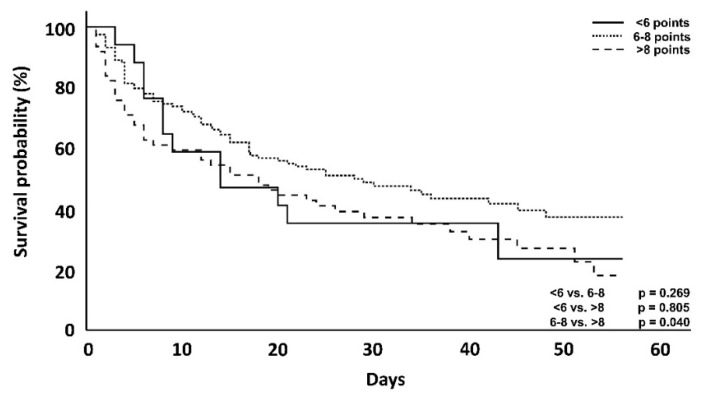
Kaplan–Meyer survival analysis at day 56 of CytoSorb treated patients. The *p*-value is via log-rank test.

**Figure 5 jcm-10-02939-f005:**
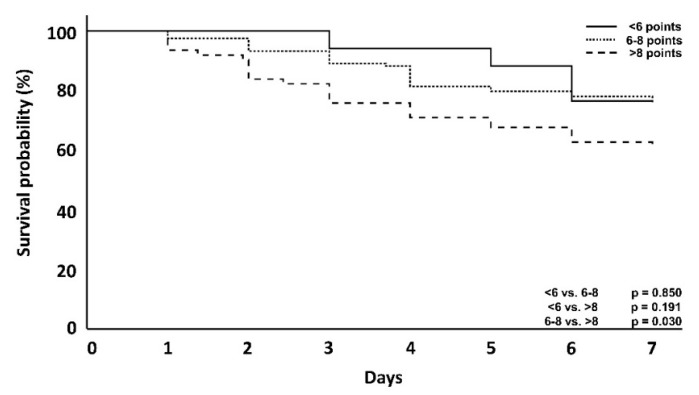
Kaplan–Meyer survival analysis at day 7 of CytoSorb treated patients. The *p*-value is via log-rank test.

**Figure 6 jcm-10-02939-f006:**
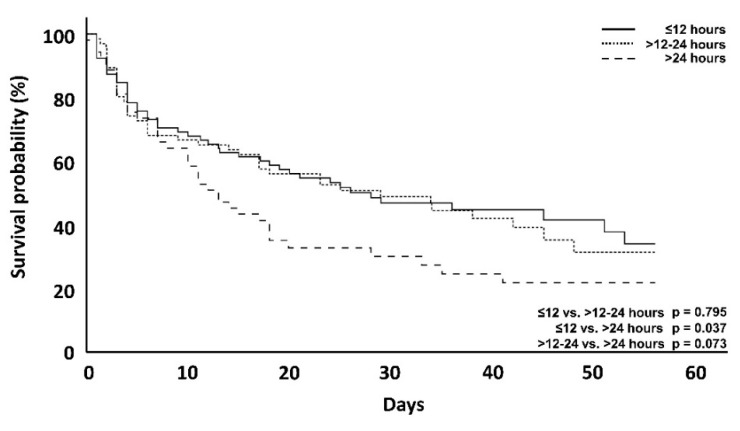
Kaplan–Meyer survival analysis of CytoSorb treated patients at day 56 with regard to different delay groups. The *p*-value is via log-rank test.

**Figure 7 jcm-10-02939-f007:**
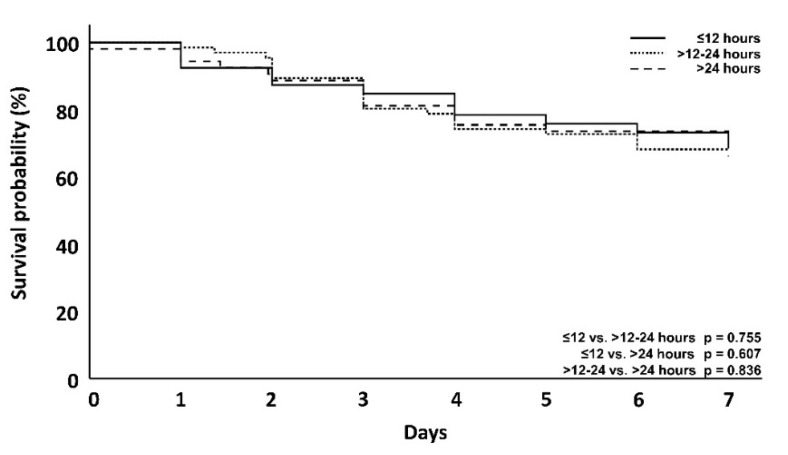
Kaplan–Meyer survival analysis of CytoSorb treated patients at day 7 with regard to different delay groups. The *p*-value is via Log-rank test.

**Table 1 jcm-10-02939-t001:** Baseline characteristics, DSS relevant parameters and outcome variables depending on score groups in the overall patient cohort. Presented are mean values ± standard deviations, frequency and percent (%) and levels of significance.

	DSS < 6 (*n* = 98)	DSS 6–8 (*n* = 294)	DSS > 8 (*n* = 110)	*p*-Value (DSS < 6 vs. DSS 6–8)	*p*-Value (DSS < 6 vs. DSS > 8)	*p*-Value (DSS 6–8 vs. DSS > 8)
Age (years)	68.5 (±12.4)	65.11 (±14.7)	66.6 (±14.4)	0.027	0.325	0.335
APACHE II (points)	35.6 (±9.2)	36.1 (±10.2)	36.9 (±9.5)	0.617	0.321	0.515
SAPS II (points)	50.2 (±13.2)	53.7 (±14.8)	62.3 (±18.6)	0.033	<0.001 *	<0.001 *
Ventilator days	7.8 (±10.1)	11.1 (±11.1)	11.7 (±16.1)	0.016 *	0.038	0.776
ICU stay (days)	13.2 (±11.9)	17.2 (±21.0)	16.4 (±19.8)	0.073	0.173	0.695
Hospital stay (days)	24.1 (±24.1)	26.3 (±31.0)	23.1 (±34.0)	0.457	0.804	0.379
ICU mortality (%)	42 (42.9%)	165 (56.1%)	71 (64.5%)	0.023	0.002 *	0.127
Hospital mortality (%)	48 (49.0%)	173 (58.8%)	79 (71.8%)	0.089	<0.001 *	0.017 *
Lactate T_0_ (mmol/L)	3.90 (±3.61)	4.65 (±3.50)	4.93 (±3.21)	0.074	0.032	0.450
Lactate T_6_ (mmol/L)	2.65 (±2.89)	3.98 (±3.25)	6.74 (±3.85)	<0.001 *	<0.001 *	<0.001 *
Norepinephrine T_0_ (µg/kg/min)	0.21 (±0.25)	0.39 (±0.41)	0.43 (±0.42)	<0.001 *	<0.001 *	0.314
Norepinephrine T_6_ (µg/kg/min)	0.20 (±0.19)	0.49 (±0.40)	0.72 (±0.36)	<0.001 *	<0.001 *	<0.001 *
Second catecholamine T_0_ (%)	3 (3.1%)	81 (27.6%)	68 (61.8%)	<0.001 *	<0.001 *	<0.001 *
Hydrocortisone T_0_ (%)	5 (5.1%)	135 (45.9%)	89 (80.9%)	<0.001 *	<0.001 *	<0.001 *
Volume bolus used (mL/kg)	62.4 (±20.4)	80.4 (±28.8)	90.0 (±31.8)	<0.001 *	<0.001 *	0.006 *
Dynamic Scoring System (points)	4.41 (±0.94)	7.04 (±0.81)	9.63 (±0.77)	<0.001 *	<0.001 *	<0.001 *

* = Statistically significant using Bonferroni-adjusted alpha critical value = 0.017.

**Table 2 jcm-10-02939-t002:** Baseline characteristics, DSS relevant parameters and outcome variables depending on score groups in the CytoSorb group. Presented are mean values ± standard deviations, frequency and percent (%) and levels of significance.

	DSS < 6 (*n* = 17)	DSS 6–8 (*n* = 118)	DSS > 8 (*n* = 63)	*p*-Value (DSS < 6 vs. DSS 6–8)	*p*-Value (DSS < 6 vs. DSS > 8)	*p*-Value (DSS 6–8 vs. DSS > 8)
Age (years)	66.8 (±10.43)	60.5 (±14.80)	64.4 (±15.76)	0.037	0.462	0.097
APACHE II (points)	34.0 (±9.29)	33.4 (±10.23)	34.6 (±10.03)	0.811	0.838	0.473
SAPS II (points)	56.2 (±18.81)	56.6 (±15.30)	65.8 (±19.85)	0.947	0.079	<0.001 *
Ventilator days	7.6 (±8.17)	13.9 (±19.05)	13.0 (±18.53)	0.022	0.247	0.773
ICU stay (days)	12.4 (±8.20)	21.4 (±25.20)	17.9 (±22.85)	0.147	0.123	0.355
Hospital stay (days)	26.1 (±33.70)	30.9 (±37.39)	25.35 (±40.39)	0.599	0.939	0.356
ICU mortality (%)	11 (64.7%)	66 (55.9%)	41 (65.1%)	0.501	0.978	0.235
Hospital mortality (%)	13 (76.5%)	67 (56.8%)	46 (73.0%)	0.122	0.774	0.032
CytoSorb therapy delay (hours)	52.6 (±30.50)	23.0 (±21.50)	18.20 (±20.57)	<0.001 *	<0.001 *	0.138
Number of CytoSorb adsorbers used (n)	2.2 (±0.77)	2.7 (±1.57)	2.7 (±1.58)	0.230	0.117	0.895
Lactate T_0_ (mmol/L)	3.12 (±3.49)	4.87 (±3.81)	5.01 (±3.26)	0.076	0.041	0.805
Lactate T_6_ (mmol/L)	2.40 (±3.02)	4.09 (±3.34)	6.60 (±3.17)	0.051	<0.001 *	<0.001 *
Norepinephrine T_0_ (µg/kg/min)	0.31 (±0.26)	0.48 (±0.50)	0.47 (±0.46)	0.036	0.169	0.906
Norepinephrine T_6_ (µg/kg/min)	0.31 (±0.25)	0.50 (±0.42)	0.75 (±0.35)	0.012 *	<0.001 *	<0.001 *
Second catecholamine T_0_ (%)	3 (17.6%)	57 (48.3%)	51 (81.0%)	0.017 *	<0.001 *	<0.001 *
Hydrocortisone T_0_ (%)	4 (23.5%)	78 (66.1%)	54 (85.7%)	<0.001 *	<0.001 *	0.003 *
Volume bolus used (mL/kg)	63.0 (±14.4)	77.4 (±27.0)	82.8 (±28.8)	0.038	0.011 *	0.261
Dynamic Scoring System (points)	4.23 (±0.97)	7.22 (±0.82)	9.84 (±0.86)	<0.001 *	<0.001 *	<0.001 *

* = Statistically significant using Bonferroni-adjusted alpha critical value = 0.017.

**Table 3 jcm-10-02939-t003:** Mortality rates in different patient groups. Presented are frequencies, percentages and levels of significance.

	**All Patients,** ** DSS < 6**	**All Patients,** **DSS 6–8**	**All Patients,** **DSS > 8**	***p*-Value** ** (<6 vs. 6–8)**	***p*-Value** **(<6 vs. >8)**	***p*-Value** ** (6–8 vs. >8)**
7-day mortality	19 (19.4%)	85 (28.9%)	42 (38.2%)	0.051	0.002 *	0.057
56-day mortality	45 (45.9%)	165 (56.1%)	76 (69.1%)	0.164	0.004 *	0.027
ICU mortality	42 (42.8%)	165 (56.1%)	71 (64.5%)	0.023	0.002 *	0.127
Hospital mortality	48 (48.9%)	173 (58.8%)	80 (72.7%)	0.089	<0.001 *	0.017 *
	**CytoSorb Patients,** ** DSS < 6**	**CytoSorb Patients,** **DSS 6–8**	**CytoSorb Patients,** **DSS > 8**	***p*-Value** ** (<6 vs. 6–8)**	***p*-Value** **(<6 vs. >8)**	***p*-Value** ** (6–8 vs. >8)**
7-day mortality	4 (23.5%)	29 (24.4%)	24 (38.7%)	0.850	0.030	0.191
56-day mortality	12 (70.5%)	66 (55.5%)	44 (71.0%)	0.269	0.040	0.805
ICU mortality	11 (64.7%)	66 (55.9%)	41 (65.0%)	0.501	0.978	0.235
Hospital mortality	13 (76.4%)	67 (56.7%)	47 (74.6%)	0.102	0.877	0.018
	**CytoSorb Therapy ** ** Delay ≤ 12 h**	**CytoSorb Therapy ** ** Delay > 12–24 h**	**CytoSorb Therapy ** ** Delay > 24 h**	***p*-Value ** ** (≤12 vs. >12–24 h)**	***p*-Value ** ** (≤12 vs. >24 h)**	***p*-Value ** ** (>12–24 vs. >24 h)**
7-day mortality	23 (29.1%)	21 (31.8%)	18 (34.0%)	0.755	0.607	0.836
56-day mortality	44 (55.7%)	39 (59.1%)	39 (73.6%)	0.795	0.037	0.073
ICU mortality	40 (50.0%)	39 (60.0%)	39 (73.5%)	0.231	0.006 *	0.123
Hospital mortality	45 (56.2%)	40 (61.5%)	42 (79.2%)	0.430	0.006 *	0.038

* = Statistically significant using Bonferroni-adjusted alpha critical value = 0.017.

**Table 4 jcm-10-02939-t004:** Baseline characteristics, DSS relevant parameters and outcome variables depending on therapy delays in the CytoSorb group. Presented are mean values ± standard deviations, frequency and percent (%) and levels of significance.

	Therapy Delay ≤12 h (*n* = 80)	Therapy Delay >12–24 h (*n* = 65)	Therapy Delay >24 h (*n* = 53)	*p*-Value (<12 vs. 12–24 h)	*p*-Value (<12 vs. >24 h)	*p*-Value (12–24 vs. >24 h)
Age (years)	62.0 (±13.72)	61.2 (±16.06)	64.1 (±15.28)	0.670	0.410	0.310
APACHE II (points)	32.7 (±9.38)	36.2 (±9.31)	32.7 (±11.53)	0.028	0.994	0.078
SAPS II (points)	64.2 (±19.26)	56.1 (±14.93)	56.7 (±16.96)	0.007 *	0.021	0.850
Ventilator days	12.9 (±18.50)	13.2 (±19.11)	13.0 (±16.93)	0.930	0.996	0.937
ICU stay (days)	20.2 (±25.80)	19.9 (±23.95)	18.0 (±19.49)	0.930	0.578	0.643
Hospital stay (days)	28.1 (±39.99)	32.1 (±42.71)	25.0 (±29.98)	0.560	0.609	0.293
ICU mortality (%)	40 (50.0%)	39 (60.0%)	39 (73.5%)	0.231	0.006 *	0.123
Hospital mortality (%)	45 (56.2%)	40 (61.5%)	42 (79.2%)	0.430	0.006 *	0.038
CytoSorb therapy delay (hours)	6.8 (±4.50)	20.4 (±3.99)	54.6 (±25.92)	<0.001 *	<0.001 *	<0.001 *
Number of CytoSorb adsorbers used (*n*)	3.03 (±1.66)	2.69 (±1.62)	2.24 (±0.97)	0.211	0.002 *	0.082
Lactate T_0_ (mmol/L)	4.73 (±3.48)	5.11 (±3.90)	4.38 (±3.55)	0.540	0.574	0.292
Lactate T_6_ (mmol/L)	5.09 (±3.63)	5.05 (±3.67)	3.80 (±3.00)	0.940	0.034	0.046
Norepinephrine T_0_ (µg/kg/min)	0.50 (±0.65)	0.45 (±0.26)	0.42 (±0.37)	0.506	0.323	0.613
Norepinephrine T_6_ (µg/kg/min)	0.64 (±0.46)	0.54 (±0.35)	0.50 (±0.40)	0.153	0.083	0.647
Second catecholamine T_0_ (%)	64 (80.0%)	25 (38.0%)	22 (41.5%)	<0.001 *	<0.001 *	0.740
Hydrocortisone T_0_ (%)	63 (78.7%)	44 (67.6%)	29 (56.6%)	0.134	0.008 *	0.235
Volume bolus used (mL/kg)	75.0 (±25.8)	81.6 (±28.8)	78.0 (±27.6)	0.122	0.510	0.444
Dynamic Scoring System (points)	8.48 (±1.53)	7.60 (±1.58)	7.01 (±2.19)	0.001 *	<0.001 *	0.093

* = Statistically significant using Bonferroni-adjusted alpha critical value = 0.017.

**Table 5 jcm-10-02939-t005:** Baseline characteristics, DSS relevant parameters and outcome variables in control + RRT patients and CytoSorb-treated individuals. Presented are mean values ± standard deviations, frequency and percentage (%) and levels of significance.

	Control + RRT Group (*n* = 69)	CytoSorb Group (*n* = 198)	*p*-Value
Age (years)	66.2 (±12.4)	62.3 (±14.9)	0.035
APACHE II (points)	39.8 (±9.6)	33.8 (±10.0)	<0.001
SAPS II (points)	56.2 (±14.8)	59.5 (±17.6)	0.144
Ventilator days	12.3 (±13.4)	13.0 (±18.2)	0.715
ICU stay (days)	19.9 (±18.1)	19.5 (±23.5)	0.882
Hospital stay (days)	30.2 (±28.9)	28.7 (±37.9)	0.703
ICU mortality (%)	44 (63.8%)	118 (59.6%)	0.540
Hospital mortality (%)	49 (71.0%)	123 (63.6%)	0.268
Lactate T_0_ (mmol/L)	4.96 (±4.28)	4.76 (±3.63)	0.738
Lactate T_6_ (mmol/L)	5.08 (±4.40)	4.73 (±3.52)	0.565
Norepinephrine T_0_ (µg/kg/min)	0.37 (±0.41)	0.46 (±0.48)	0.142
Norepinephrine T_6_ (µg/kg/min)	0.55 (±0.44)	0.57 (±0.41)	0.783
Second catecholamine T_0_ (%)	17 (24.6%)	111 (56.1%)	<0.001
Hydrocortisone T_0_ (%)	37 (53.6%)	136 (68.7%)	0.024
Volume bolus used (mL/kg)	75.6 (±30.0)	78.0 (±27.6)	0.537
Dynamic Scoring System (points)	7.20 (±1.68)	7.80 (±1.82)	0.014

**Table 6 jcm-10-02939-t006:** Association between DSS score, therapy delay and outcome between control + RRT vs. CytoSorb group. Please note that groups containing not enough patients for a reliable statistical evaluation were classified as not applicable (n.a.); therefore, no analysis was performed in these groups. Presented are frequency and percentage (%) and levels of significance.

	ICU Mortality			Hospital Mortality		
	Control + RRT Group (*n* = 69)	CytoSorb Group (*n* = 198)	*p*-Value	Control + RRT Group (*n* = 69)	CytoSorb Group (*n* = 198)	*p*-Value
DSS <6 points, CytoSorb delay ≤12 h	n.a.	n.a.	n.a.	n.a.	n.a.	n.a.
DSS 6–8 points, CytoSorb delay ≤12 h	27 (58.7%)	21 (47.7%)	0.303	30 (65.2%)	21 (47.7%)	0.096
DSS >8 points, CytoSorb delay ≤12 h	12 (92.3%)	17 (51.5%)	0.009 *	12 (92.3%)	21 (63.6%)	0.053
DSS <6 points, CytoSorb delay >12–24 h	n.a.	n.a.	n.a.	n.a.	n.a.	n.a.
DSS 6–8 points, CytoSorb delay >12–24 h	27 (58.7%)	25 (54.3%)	0.678	30 (65.2%)	26 (56.5%)	0.398
DSS >8 points, CytoSorb delay >12–24 h	n.a.	n.a.	n.a.	n.a.	n.a.	n.a.
DSS <6 points, CytoSorb delay >24 h	n.a.	n.a.	n.a.	n.a.	n.a.	n.a.
DSS 6–8 points, CytoSorb delay >24 h	27 (58.7%)	19 (70.3%)	0.318	30 (65.2%)	19 (70.3%)	0.654
DSS >8 points, CytoSorb delay >24 h	n.a.	n.a.	n.a.	n.a.	n.a.	n.a.

* = Statistically significant using Bonferroni-adjusted alpha critical value = 0.017.

**Table 7 jcm-10-02939-t007:** Multivariate logistic regression results for predictors of day 56 survival.

Predictor Variable	Odds Ratio	95% Confidence Interval	*p*-Value
CytoSorb Therapy (Yes/No)	0.552	0.275, 1.108	0.095
DSS Score	1.237	1.106, 1.383	<0.001
Therapy Delay (h)	1.015	1.001, 1.030	0.034
RRT (Yes/No)	1.795	0.991, 3.252	0.054
Age (Years)	1.037	1.023, 1.052	<0.001

## Data Availability

The datasets used and/or analyzed during the current study available from the corresponding author on reasonable request.

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
