# Peer review of "First Evaluation of a New Dynamic Scoring System Intended to Support Prescription of Adjuvant CytoSorb Hemoadsorption Therapy in Patients with Septic Shock"

_jcm, 2021, doi:10.3390/jcm10132939_

Round 1

Reviewer 1 Report

No additional comments.

Author Response

We thank Reviewer 1 for his feedback.

Reviewer 2 Report

Kogelmann and colleagues exhibited a new scoring system possibly useful to pre-diagnose pathologic advancements of septic shock patients in this study. Throughout the revision, some parts of the manuscript have been addressed; however, the data are still required to be modified. The authors should improve the quality of all the Figures.

  • All the Figures in current form need to be improved because they are unapparent and blurry. All the graphs of Figures 2, 3, 4, 5, 6, and 7 need to have apparently visible and larger letters. All the graphs do not contain any marks indicating any statistical significance although the captions explain it. It will be better to put marks (*) inside the graphs as well. Also, please explain more about which cohorts were compared.

Author Response

Point-by-point response to the reviewer’s comments

Kogelmann and colleagues exhibited a new scoring system possibly useful to pre-diagnose pathologic advancements of septic shock patients in this study. Throughout the revision, some parts of the manuscript have been addressed; however, the data are still required to be modified. The authors should improve the quality of all the Figures.

All the Figures in current form need to be improved because they are unapparent and blurry.

All figures have been improved and are now provided in 1200 dpi.

All the graphs of Figures 2, 3, 4, 5, 6, and 7 need to have apparently visible and larger letters.

Letters in Figures 2, 3, 4, 5, 6, and 7 are displayed in a larger size now.

All the graphs do not contain any marks indicating any statistical significance although the captions explain it. It will be better to put marks (*) inside the graphs as well.

Asterisks (*) have been added to the graphs where applicable (Figure 2 and 3). In the other graphs (Figures 4-7), there is no statistical significance for none of the comparisons. Additionally, we have also added new words to the legends on each figure to explain what is significant, and what is not significant.

Also, please explain more about which cohorts were compared.

As explained in the former point, we have added new words to the legends on each figure to explain what is significant, and what is not significant while group/cohort comparisons are now displayed in each single figure (Figure 2-7).

This manuscript is a resubmission of an earlier submission. The following is a list of the peer review reports and author responses from that submission.

Round 1

Reviewer 1 Report

See document attached. 

Author Response

The authors created a dynamic scoring system (DSS) to assess patients with early septic shock and performed a first evaluation of the system in this patient population. They were able to detect populations with a distinct mortality pattern based on the DSS. Furthermore, the authors showed that an early start of CytoSorb therapy was associated with significantly improved survival.

  1. Please provide definitions for abbreviations in parentheses the first time they appear. The definitions for PCT and CRP on page 2 in line 86 and for the disease severity scores APACHE II and SAPS 2 on page 2 in line 91 are missing. The definition for NE in Figure 1 is also missing.

We thank the reviewer for this important hint and have provided definitions for abbreviations in parentheses the first time they appear both in the text and in Figure 1.

  1. In the statistical section you write that data are presented as mean ± standard deviation or median but in your tables and figures you don’t provide which value is presented, the median or the mean. Please indicate in the figure legend and table caption which values are presented.

We have clarified these issues as requested both in the figure legends and in the table captions.

  1. Error bars in Figure 8 are missing.

This is an important issue that is raised by the reviewer and we came across this several times during the preparation of the data and the manuscript. After consultation of several internal but also external statisticians we came to the conclusion, that reporting of standard deviations or confidence intervals (both being an estimate of the uncertainty regarding sample proportions) in our case of a reported mortality as a percentage is not feasible. This is due to the fact that assessment of mortality belongs to descriptive statistics and is a binary subject, as there are only two possible outcomes i.e. survival no = 0 vs. survival yes = 1. Therefore measures of variation for a percentage value seems inappropriate. However, we have updated the figure by improving comprehensibility both in the figure itself as well as in the figure legend.

  1. Please provide the number of included patients (n) in Figure 8 and Table 4.

We thank the reviewers for having spotted these inconsistences and gaps, which were now corrected or filled respectively with the new version.

  1. The authors reported that the inflammatory parameters (PCT, CRP) were included in the secondary objectives, but these parameters did not appear in the further course of this study.

Provide the inflammatory parameters in the different score groups. Were other inflammation markers such as cytokines also measured? Would it be possible that the inflammation parameters and the possibly increased cytokine production have an influence on the high mortality score group 1 in the CytoSorb group after 7 and 56 days?

Thank you for making us aware of having these parameters left out in the first place and also for coming up with a reasonable hypothesis for explaining the higher mortality in group 1. To deal with the reviewers query, we have provided a table showing plasma levels of CRP and PCT and comparing them in relation to the DSS but also with regard to time delay. Unfortunately, however, analysis of inflammatory parameters does not provide any explanation. In the group including the entire patient cohort with a DSS <6 points as well as in the CytoSorb group with DSS <6 points, the PCT is the lowest, which does not point towards a foreseeable cytokine storm. Apart from that (groups < 6), there are no other significant differences when comparing inflammatory parameters against individual score groups.

We therefore included the following statement into the results section:

“Analysis of inflammatory parameters did not show any correlation between score groups apart from patients with a DSS <6 points had the lowest PCT levels.”

Importantly, we did not measure any cytokines in our setting as these were not routinely available in all participating centers. However, this was exactly the purpose of our study to provide an easy to implement scoring system with predominantly clinical parameters.

  1. Line 240-244: With one exception, the correlation between higher scores and mortality was the same in the CytoSorb treated patient population (Table 2, Figure 4, Figure 5), as CytoSorb treated patients with DSS >6 failed to show this correlation. Similar findings were observed by Ferreira and colleagues in an analysis of outcomes related to changes in the SOFA score (23).

Did you analyze the correlation between your DSS and the SOFA score?

A comparison between SOFA score and DSS would surely have been very interesting, so we thank the reviewers for this input. Undoubtedly, the SOFA score provides a broader picture of the patient’s clinical status and should definitely not be ignored when deciding for or against initiation of CytoSorb therapy in a patient. However, as the primary therapeutic effect of CytoSorb therapy (according to our personal experiences and the published literature) is hemodynamic stabilization and shock control in refractory septic (and vasopolegic) shock states, we wanted to specifically address this clinical condition by assessing parameters such as catecholamine demand, volume requirements, lactate as well as their development in the early phase (first 6 hours) with our scoring system.

On the contrary, the predictive value for mortality using the SOFA score is validated for changes in the first 48-96 hours. With regard to the values necessary to acquire the SOFA score (i.e. bilirubin, thrombocytes, creatinine, P/F ratio), it becomes obvious, that a clear change of these (slow reacting) laboratory values in the first 6 hours is unlikely to occur and thus a change in the SOFA as a whole also is. This is why we have not included the SOFA score nor any comparisons in our approach. Taking Ferreira et al. as a sort of reference is derived from the fact, that patients with a lower risk score also had a worse outcome. Lastly, an evaluation of the SOFA score is not easily possible with our data set, as most of the necessary data (Glasgow-Coma-Scale score, bilirubin, thrombocytes, PF ratio) were not collected at the treating centers. However, this can certainly be part of future investigations for which our manuscript might serve as a good base.

Have you the same outcomes when you correlate the SOFA-Score of your patients with the mortality?

As explained above, this would have been a very nice further analysis, which however was not possible to be done at this stage, but will be considered as a very valuable input for upcoming analysis of our database, which have already started to be discussed internally.

Is a low SOFA score equal to a low DSS?

As the SOFA score for example doesn’t reflect changes on NE needs beyond 0.1 µg/kg/min in a higher scoring, but does on the other hand involve many other important clinical information, which are ignored in our scoring system, there should be certain differences when comparing the two. Unfortunately and as explained earlier, we were not able to perform such analysis in the current manuscript.

Reviewer 2 Report

Authors have done a great job in explaining the new dynamic scoring system that they developed to help doctors to prescribe the auxiliary CytoSorb hemoadsorption therapy. The manuscript was pleasant to read.

However, after carefully reading the manuscript, I suggest authors to increase the quality of all figures. Majority of figures were very difficult to read and understand. It would be helpful to readers to comprehend the data easily if the quality of figures is enhanced.

Author Response

Authors have done a great job in explaining the new dynamic scoring system that they developed to help doctors to prescribe the auxiliary CytoSorb hemoadsorption therapy. The manuscript was pleasant to read.
However, after carefully reading the manuscript, I suggest authors to increase the quality of all figures. Majority of figures were very difficult to read and understand. It would be helpful to readers to comprehend the data easily if the quality of figures is enhanced.

We thank the reviewer for his query and have provided all graphs in good resolution (600 dpi) and tried to optimize the figures for better comprehension e.g. by adding a table into each Kaplan-Meier-Survival curve showing p values for the various comparisons as well as titles for Figure 8. Furthermore, we updated figure legends by providing more data for better comprehensibility.

Reviewer 3 Report

Kogelmann and colleagues have shown a newly devised scoring system to potentially pre-diagnose pathologic advancements of septic shock patients in this study. The data shown in this article exhibited the efficacy and requirement for additional administration of CytoSorb via analyzing retrospectively documented data, which is evident that the preceding treatment of CytoSorb represents a better remedy. This manuscript is well-written and can get influential. Particularly, it is envisaged that this newly devised scoring method provides any prognostic evidence instrumental in treating early sepsis patients. Considering its clinical relevancy and importance, I recommend this article for positive consideration as to publishing in Journal of Clinical Medicine as an Article. I have only some of minor comments which can be promptly addressed by the authors at convenience.

  • There are some omissions of marks after authors’ names (4 and *) in the front page.
  • Figures 2 to 7 need to be improved because the graphs and the inside words are blurry. Also, it is unclear which cohorts were mutually compared in terms of p values and arrows indicated in Figures 4, 5, and 7, two out of three cohorts (<6, 6-8, and >8)? otherwise or all three cohorts?.
  • It would be better to describe Kaplan-Meier method used in current study, as a separate section in Methods for general readership.
  • There some bracket errors for indicating reference numbers (such as lines 254, 256, 258…). Need to check them in all text.
  • a (alpha) is not necessary in line 140. Just write “statistically significant”.

Author Response

Kogelmann and colleagues have shown a newly devised scoring system to potentially pre-diagnose pathologic advancements of septic shock patients in this study. The data shown in this article exhibited the efficacy and requirement for additional administration of CytoSorb via analyzing retrospectively documented data, which is evident that the preceding treatment of CytoSorb represents a better remedy. This manuscript is well-written and can get influential. Particularly, it is envisaged that this newly devised scoring method provides any prognostic evidence instrumental in treating early sepsis patients. Considering its clinical relevancy and importance, I recommend this article for positive consideration as to publishing in Journal of Clinical Medicine as an Article. I have only some of minor comments which can be promptly addressed by the authors at convenience.

There are some omissions of marks after authors’ names (4 and *) in the front page.

We thank the reviewer for this important hint. Both issues have been fixed accordingly on the front page.

Figures 2 to 7 need to be improved because the graphs and the inside words are blurry. Also, it is unclear which cohorts were mutually compared in terms of p values and arrows indicated in Figures 4, 5, and 7, two out of three cohorts (<6, 6-8, and >8)? otherwise or all three cohorts?.

We have improved overall appearance of all Kaplan-Meier-Survival curves, added a table into each graph showing p values for the various comparisons and have provided all graphs in good resolution.

It would be better to describe Kaplan-Meier method used in current study, as a separate section in Methods.

As suggested by the reviewer we have categorized the Kaplan-Meier method into a separate section in the Methods chapter and hope that this improves general readership.

There some bracket errors for indicating reference numbers (such as lines 254, 256, 258…). Need to check them all.

We thank the reviewer for this important hint. This has all been fixed accordingly in the entire text.

a (alpha) is not necessary in line 140. Just write “statistically significant”.

We thank the reviewer and have adjusted that issue accordingly.